# The PRU: The Solution for Whom?

**Johan Malmqvist** [1,2]

1    Faculty of Social Sciences, Linnaeus University, 351 95 Växjö, Sweden; johan.malmqvist@lnu.se
2    School of Education and Communication, Jönköping University, 5. Box 1026, 551 11 Jönköping, Sweden

**Abstract:** In Sweden, pupil referral units (PRUs) have been recommended by the government, suggesting that "inclusion has gone too far". This governmental recommendation is not based on research focusing on PRUs, as such research is sparse. Furthermore, there has been a lack of evaluations of the efficacy of PRUs, and no national evaluations of such provision have been undertaken. Furthermore, more attention must be paid to PRU students' own perspectives and experiences as we lack knowledge of their needs and situation. This study aimed to investigate how educational needs have been and should be addressed in one PRU according to nine stakeholder groups, for example, current students, former students, parents, school staff, and various groups of people who, in their work, were responsible for deciding about the PRU (e.g., chief education officers or politicians) or supporting the PRU (e.g., school healthcare unit staff). Comparative analysis of all groups' perceptions considered similarities and differences of views of this topic. Preliminary results indicate substantial between- and within-group variation concerning the purpose of the PRU and uncertainty about educational quality, partly due to insufficient documentation. Some students described a "Catch-22": having been told to catch up educationally with peers and that PRU placement would help in this, they were disappointed, as the emphasis on non-educational practices impeded catching up.

**Keywords:** PRU; special needs education; special education; segregation; inclusion; inclusive education; stakeholder

## 1. Introduction

For about twenty years, Swedish schools have struggled with poor results in international tests, such as PISA, and increased inequity. This has prompted a long-running debate about the causes of the current state. In parallel, the social order in schools and the efficacy of inclusion have been questioned. The establishment of Pupil Referral Units (PRUs) has been recommended by the government, as it is suggested that "inclusion has gone too far" [1]. Students who experience severe school difficulties concerning behavioural issues, such as students displaying emotional and behavioural difficulties, are the main target of PRU schooling. In some cases, PRUs address students with various causes of their school difficulties, whereas other PRUs target only students with certain disabilities, such as speech impairments, or certain diagnoses, such as ADHD. PRUs are often small with under 10 students; they follow the national curriculum and are often governed by municipalities, such as the PRU investigated here. Placement in a municipal PRU is often unavailable until it has been established that the regular school, despite extensive special needs support from a central municipal unit, cannot adequately address the student's needs. This depends partly on the inclusive education capacities of regular schools, as even regular schools that are similar in most other ways may differ greatly in this regard [2].

To understand the demand for and growth in PRUs in Sweden, it is important to understand the presence of private companies that administer PRUs as independent schools. Here, "PRU" refers to schools that restrict enrolment to students needing special support, irrespective of whether these schools are private or owned by municipalities. PRUs are to offer temporary school placement, as their students should eventually re-enter regular

schools; however, this is neither a realistic nor desirable aim for all PRU students, as stated by a recent school commission [3]. The quotation above, which states that inclusion has gone too far, shows that there is party-political interest in where students who encounter school difficulties will receive their education. In 2010, the previous government reformed educational legislation to make it legal for independent schools to restrict their enrolment to students needing special support, something they had done without permission for several years [3]. According to Magnusson, some private companies are clustering pupils needing special support in PRUs as a marketing idea, and have renewed the idea of "special schools" for students needing special support [4,5]. In an educational system based on school choice, many such schools have claimed to have expertise and competence to provide good-quality education for students failed by mainstream schools. This has led to a less inclusive education system and is partially a consequence of the school choice idea, in which parental/individual preferences and a client focus have been prioritized [4].

To grasp the complexity of the Swedish school system, it should be noted that the School Inspectorate has long claimed that PRUs governed by municipalities contravene legislation, leading to a decreased number of municipal PRUs. However, not all 290 municipalities agree with this interpretation of educational legislation. In 2017, the Supreme Administrative Court established that municipally run PRUs are legal, and several new PRUs have since been established. The municipal desire to establish PRUs can partly be regarded as a consequence of the development of the Swedish market-driven educational system. When students and/or their legal guardians choose independent schools specializing in students with a medical diagnosis, such as ADHD, municipalities may incur substantial costs. The establishment of municipal PRUs specializing in students with diagnoses such as ASD and ADHD may indicate a strategy to reduce such costs.

As described above, current national policy is diametrically opposed to the previous policy of decreasing the number of segregated educational settings. This about-face follows the pattern of political standpoints in Sweden swinging from one pole to another regarding inclusion and segregation. Another example can be seen in teacher training policy, with two opposite positions being proposed concerning special pedagogical policy between 1999 and 2008 [6]. Today, it is obvious that the previous Swedish policy of inclusive education has been challenged, being partially replaced with one of segregation. The approval of privately owned PRUs, followed by recommendations to increase the use of PRUs by municipalities, indicates that the policy of segregation in education has gained strength in Sweden. Notably, in instituting these changes regarding PRUs and other segregation measures, the Swedish government has emphasized that schools must base their work on science, as stipulated in educational legislation. However, the government recommended an increase in PRUs without scientific support [3]. Instead, this segregation measure was based on agreements between political parties in January 2019, when parliamentary deadlock was resolved with an agreement covering 73 issues negotiated by two non-governmental political parties bargaining with the coalition government. The agreement decided that the Swedish National Agency for Education should investigate how to make it easier for municipalities to establish new PRUs.

The establishment of new PRUs has started to affect the school conditions of some learners. These learners often have underlying socioeconomic disadvantages, and many also have neuropsychiatric diagnoses. They are learners who run a high risk of being marginalized as adults as well, especially if they are locked into a segregated pathway through the education system [7]. A recent governmental commission stressed that school principals view segregated schooling as disadvantageous for learners [3]. According to the commission's review of PRU research and evaluations, we lack knowledge of PRU students' own perspectives on and experience of PRU schooling. This study was accordingly intended to build such knowledge and make relevant comparisons by also examining other groups affected by PRUs and groups that have power to affect the development of PRUs.

The primary aim of this study was to investigate stakeholders' experiences of work in one PRU with a secondary aim of comparing students' experiences with those of other

stakeholders. It was considered important to initiate discussion of the role of PRUs in the school system, discussion substantially informed by knowledge of students´ experiences. This is in agreement with recent Swedish legislation implementing the United Nations Convention on the Rights of the Child.

The research questions focusing on the stakeholder groups' perceptions were:

1.  What are the underlying problems that make PRU placements necessary for some students?
2.  What are the relevant PRU policies and their main objectives?
3.  What support does the PRU provide?
4.  What are the outcomes of the PRU according to stakeholders?
5.  What experiences have stakeholders had of PRU placements?
6.  What are the alternatives to the PRU and how are they perceived?

## 2. Prior Research

A literature review identified few studies investigating PRU students' views and experiences regarding their school situation and schooling. A recent literature review by the National Agency for Special Needs Education and Schools (SPSM) [8] entitled "A PRU knowledge base" (translated from Swedish) found only four publications on PRUs, three of which were written by the present author. The SPSM's literature review found that students with an ADHD or ASD diagnosis are more often segregated to PRUs than other students, based on a total population study by Malmqvist and Nilholm [9]. Furthermore, there is greater demand for PRU placements for these two groups of students than for other groups of students. These demands are mainly voiced by parents of children with these diagnoses, parents of children whose peers have these diagnoses, interest groups, and teachers. Based on this demand, many independent schools (i.e., "free schools") have adopted a PRU profile, according to a mapping report by the Swedish Agency for Education [10]. According to the SPSM [8], this indicates increasing segregation in the Swedish educational system. The Swedish Association of Local Authorities and Regions has argued that municipal PRU placements can be viewed as a long-term strategy to foster inclusion; however, according to the review's conclusion, this claim is unsupported. Furthermore, and as reported by the Swedish Agency for Education report [10], PRU students often want to return to regular schools but do not know how to do this [8].

PRUs are also used in other countries, such as the UK, where the objective of sometimes using PRUs for excluded students [11] differs from the objective in Sweden, as "deep exclusion" [12] does not exist in the latter. According to Hart [13], PRUs were introduced in England and Wales in 1994 with the aim of taking children "off track" for a period before helping them back on track for successful reintegration into mainstream schools [11,13]. In Hart's study [13], six children and the staff of one PRU were interviewed. The children, who were known to Social Services, were aged 9–13 years and subject to several risk factors. In investigating potential protective factors identified by children and staff, Hart found three key factors: attachment relationships, adult support, and personalized learning. The reality is that most PRUs lack these factors, as there are problems with poor-quality education provision contributing to academic underachievement and negative life trajectories [11]. In comparing the views of primary- and secondary-education PRU students, Jalali et al. [11] investigated how the students attributed their difficulties. The researchers found small differences between the two groups, but the secondary-education PRU population showed greater awareness of environmental factors influencing their situation, such as home and teacher relationships. The researchers also investigated the students' views of reintegration into regular schools. Whereas most of the younger students said that they wanted to return to mainstream education, the older students largely wanted to opt out. The views of PRU versus mainstream school staff regarding reintegration were investigated by Lawrence [14] using focus groups. Lawrence found that it was crucial that the students should want to return to mainstream schools if the reintegration was to succeed. Another

factor promoting successful reintegration was that the parents were engaged in their child and her/his education.

The educational quality in regular schools corresponds to the need for PRU placements. A review of students with emotional and behavioural difficulties (EBD) in mainstream schools that focused on teachers' attitudes toward including students with EBD in mainstream settings [15] is relevant here. PRUs' descriptions of their target groups often describe these students as having EBD. In addition, these students often have neuropsychiatric diagnoses [2,9,16], and special education classes have been established for diagnoses such as ADHD [14]. Gidlund [15] found 15 studies from 15 countries that considered teacher attitudes and multiple student groups. In ten of these studies, students with EBD were considered by teachers to be the most difficult group to include in mainstream settings. The teachers' attitudes were dependent on the nature of the disability, teaching experience and training, and the availability of support services. Gidlund also emphasized that cultural norms affect the attitudes toward students and that there is a need for further research into what support mainstream teachers need.

To sum up, studies of PRUs in Sweden are lacking and studies of students' experiences of their schooling in PRUs could not be found. Of the limited number of international studies of PRUs, few include reports on PRU students' experiences. Of course, a large body of research explores other types of segregated educational settings for students who experience school difficulties or have disabilities. The present review, however, is restricted to research on PRUs, as they represent a specific educational setting intended to return students to regular schools after a short period. This literature review underlines that the ongoing establishment of PRUs in Sweden has been authorized and supported without a sound basis of research knowledge.

### 3. A Framework of Stakeholder Views of Key Issues in Educational Settings Contributed by the IRIS Model

Stakeholder theory was chosen to collate views and opinions as it provided a theoretical tool with which to investigate and compare the experiences and views of nine stakeholder groups participating in a study of a PRU (cf. [17]). With this theoretical tool, originally described in his seminal 1984 work [18], Freeman advised organizations, for instance, educational organizations, to be aware of different stakeholder groups of importance to achieving improved performance [17]. According to Fassin ([19], p. 116), referring to Freeman's [19] classical definition, a stakeholder is a person or group who "can affect or is affected by the achievement of an organization's objectives".

A central foundation of stakeholder theory is that organizations should be managed in consideration of all who are affected by them [17]. Who are considered stakeholders depends partly on the kind of organization being examined. How to delimit who, or what groups, are identified as stakeholders has been debated [20]. In stakeholder research, some are categorized as primary stakeholders. In this study, the PRU students are considered primary stakeholders, as they have their school placements at the PRU. Other stakeholders, including students in regular schools, are considered to belong to the category of secondary stakeholders, mainly because they are in no way involved in activities at the PRU, and their schoolwork does not directly influence the PRU. These students may indirectly experience gains or losses due to the presence of a PRU in their municipality. For instance, the presence of a PRU may influence what students at a school are transferred to a PRU. This may happen if regular school students complain about the actions or behaviour of a student, which may lead to the student being segregated in a PRU. There are a number of other secondary stakeholder groups, such as the parents of students in regular schools and the teachers at these schools, who were not part of this study. The primary stakeholder groups in this study are stakeholders who had work tasks/assignments related to managing the PRU, were current pupils, or were legal guardians of pupils attending the PRU when the data were collected. In other words, the first group supplied schooling services, while the second and third groups can be viewed as service recipients ([20], p. 453, Figure 6). This is in line with Freeman's definition of a stakeholder as "any group or individual who can

affect or is affected by the achievement of the organization's objectives" ([17], p. 1160). A third strand of stakeholder theory addresses outcomes, or performance, based on how the organization works and is referred to as instrumental stakeholder theory [17]. Hence, the outcomes that the PRU is regarded as supporting, as well as the outcomes the stakeholder groups experience, will be interrogated here. However, what precedes the outcomes of the PRU is of equal importance for this study. Hence, stakeholder theory has been combined with the Inclusive Research in Irish Schools (IRIS) model.

The IRIS model [21] was developed from the findings of a literature review conducted as part of the IRIS research project; it was eventually elaborated on and applied in the IRIS empirical research study. In the present study, the experiences and views of the stakeholders are related to the four areas of the IRIS model. These areas are shown in Figure 1 and defined as policy, provision, experiences, and outcomes. In this study, the model was used as a screen to sort the empirical findings. The four IRIS model areas cover general educational issues, but not specific aspects of the PRU. The model has an inherent logic, a logical pathway, that was applied here. This pathway was formulated as and guided by the following questions, which served as research questions for this study:

1. What policy has been formulated?
2. What educational provision are administered to achieve the policy objective/objectives?
3. What are the outcomes of the administered educational provision?
4. What are the different stakeholder groups' experiences of the implemented policy in terms of policy, educational provision, and outcomes?

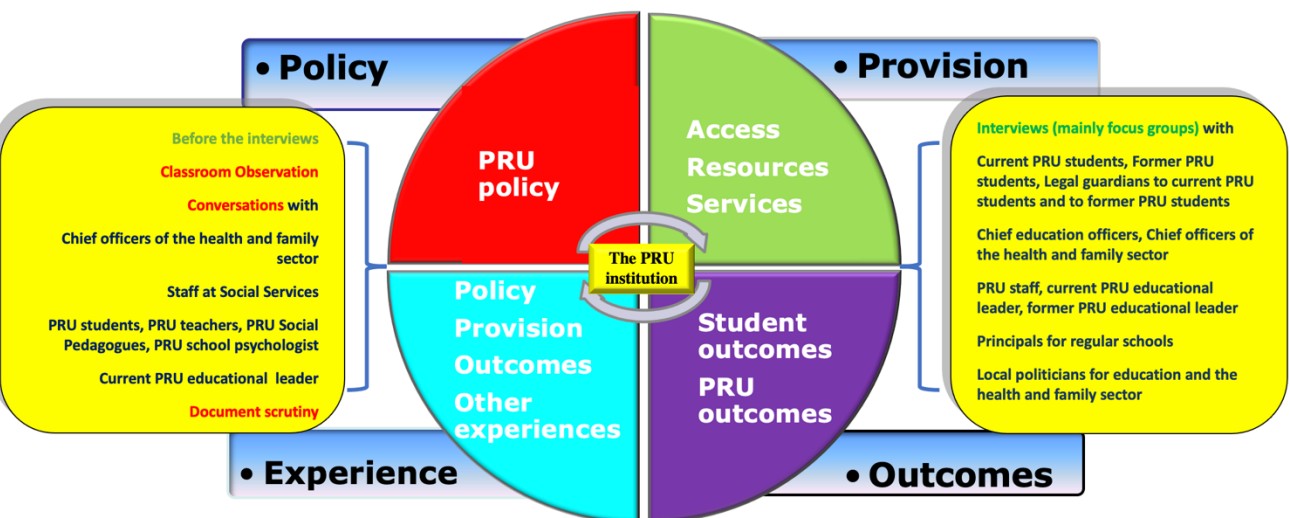

**Figure 1.** An adaption of the IRIS case study model that has been used in analysing stakeholder groups' perceptions about a pupil referral unit (PRU). Reprinted with permission from ref. [21]. Copyright year 2015, Copyright owners, Richard Rose and Michael Shevlin.

Two additional areas were also investigated and formulated as research questions: one area concerned who the prospective PRU students will be, and the other concerned whether there are alternatives to PRU placement, as described above.

## 4. Method

The study was designed as a stakeholder study based on the guidelines from the municipality that commissioned the study. One objective was to collect data from several groups representing different perspectives: students, legal guardians, professional groups whose work in some way concerns the PRU, and politicians. The consequence of this objective was that most, but not all, of the interviewees were primary stakeholders [17].

The data collection was conducted in 2011 framed within the same educational legislation that is present today [3]. A thorough analysis based on transcribed audial recordings of this data has not been conducted before, and there is no research article based on this data that has ever been published.

### 4.1. Participants

All participants were formally invited by the municipality to participate in a research-based evaluation of the PRU; it was stated that the empirical findings would also be part of a research study. Participant consent was required, and all participants were informed that participation was voluntary and that they were free to end their participation at any time. Written consent was obtained from the legal guardians.

In this study, the researcher informed the municipality who to contact as interviewees; of the 44 invited participants, 37 agreed to participate (Figure 2). Students who had their school placement at the PRU were one stakeholder group (five of the PRU students enrolled were asked to participate), and legal guardians of former students were another. Most of them were legal guardians to students who were part of the focus group that consisted of former students. Several interviewees were chosen by the researcher based on their position as municipality employees. They were working as chief education officers, chief officers of the health and family sector, principals of regular schools, and PRU staff. The PRU staff consisted of one educational leader who also taught, three teachers, one leisure time pedagogue, and two social pedagogues. The latter three also taught to some extent. However, the leisure time pedagogue mainly worked with leisure time activities, whereas the two social pedagogues worked, for instance, with the social interaction in the PRU and with supporting PRU students' families. The politicians responsible for young people and their schooling were another stakeholder group. All these interviewees were considered primary stakeholders.

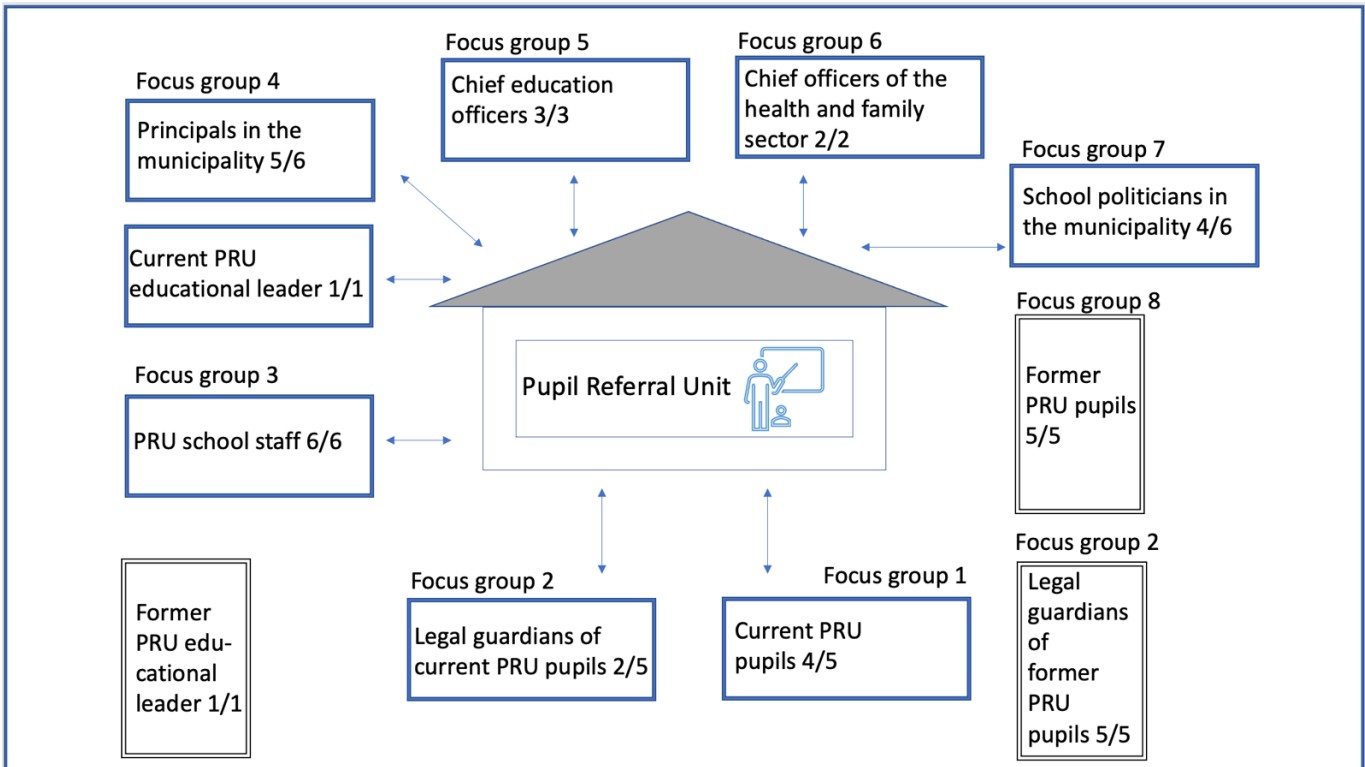

**Figure 2.** Stakeholder groups and participants in each of the eight focus group interviews and number of interviewees/number of invited participants. Primary stakeholders in blue boxes; secondary stakeholders in black boxes.

The situation differed for two other groups of stakeholders, i.e., former PRU students and legal guardians of former students, who are regarded as secondary stakeholders in this study. The municipal contact person for the project was asked by the researcher to recruit former students and their legal guardians, which was difficult for this person. Eventually, four legal guardians, including the two guardians of one former student, consented to participate in the study. They were recruited without any contact with the researcher, who received no information about them before the focus group interviews. It turned out that three of these legal guardians had been persistent in persuading the municipality to provide PRU placements for their children. Two of the legal guardians had in fact strongly pressured the municipality to establish the PRU. It is unlikely that most legal guardians whose children attended the PRU resembled these three, so in this sense, the selection of legal guardians and their children was skewed/biased. Their and their children's retrospective views were nevertheless seen as important for the study. Furthermore, the individual interview with the former educational leader of the PRU added a retrospective view of the PRU, whereas the individual interview with the current educational leader supplied an updated view. These two experienced PRU leaders were not invited to participate in the focus group interviews; rather, their views and experiences were documented during the individual interviews only. They were described as principals by some interviewees but they did not have the authority to make decisions as principals can.

### 4.2. Observations

Non-participant observations were conducted for two full school days. The observations were conducted before the interviews to contribute to a holistic perspective on the students' school situation and to better understand what was described during the focus group interviews. This meant that use of the observations depended on the nature of the research project, in which observational data mainly served to enhance the quality of the extensive interview data and where the observer acted as a witness [22].

The most attention was paid to each student's school situation, how the teaching and other school activities were organized and realized, the extent to which the students were involved in one-to-one teaching separate from the other students in their own rooms or together with other students, and social interactions between students or between students and school staff. Field notes were taken for documentation. The observational findings are not reported separately in the "Results" section as they mainly served to enhance the quality of the interview data. For example, the observation findings clarified for me, as the interviewer, certain crucial matters that were important to understand in the interviews: the relationships and, not least, the tensions between current PRU students; the types of events that occurred during regular school days; and the comments of staff interviewees who had different roles and educational backgrounds. Additionally, informal conversations during the observation days provided additional understanding, as I could ask staff about the daily routines that I was observing.

### 4.3. Interviews

The individual interviews with past and current leaders were semi-structured, and all were digitally recorded. The group interviews were also digitally recorded but differed in character from the individual ones. Whereas the individual interviews were conversations with a more or less continuous flow of words, in the focus group interviews, the interviewees were given time to reflect and to formulate their answers, resulting in frequent pauses in speech. There were also some intense discussions, for example, due to disagreement about the PRU and responsibility for students with special needs. The focus group interviews also involved individual marking and rating activities (see below), resulting in the interviewees being quiet for extended periods.

The focus group interviews had two parts. The first part was a group interview using focus group interview technique to some extent. This was followed by a focus group interview in which qualitative as well as quantitative data were collected. The format

of this focus group interview was developed by Obert and Forsell [23]. The data were digitally recorded, and the interviewees completed Excel spreadsheets in which they rated their group's formulated statements. One question was posed to all groups: What do you believe is important in order to make the PRU work well? This question was written on posters around a conference table, visible to all. The researcher sat near the end of the table with a computer connected to a projector, which projected an open Excel spreadsheet on a screen visible to all interviewees. The interviewees continued to describe what factors they considered important until they ran out of new ideas, which could be viewed as a type of saturation. All responses were immediately written down in the projected Excel spreadsheet and were revised until the responding interviewees were satisfied with the formulation. During this phase, the interviewees described the PRU at length and discussed their responses. Many clarifications were made, prompted by the researcher and other interviewees. There was no intention that the interviewees should agree on what was important. On the contrary, the interviewees were encouraged to come up with different answers based on what each of them found important. The number of factors ranged from eight (from the current PRU pupils) to 23 (from the PRU staff).

In the next phase, individual interviewees in each group were asked to rate what factors were most important. First, the interviewees were all handed pencils and printed Excel spreadsheets containing all the answers from their own group. The interviewees individually selected one third of the answers regarded as most important on their Excel spreadsheets. The PRU staff, for example, marked eight answers each, approximately a third of the answers from that group. After that, the marked-up spreadsheets were returned to the researcher. The researcher then provided the interviewees with another unmarked copy of the Excel spreadsheet containing all the answers. For example, the PRU staff again received the Excel spreadsheet with 23 answers; this time, they rated all 23 answers on a scale of 1–5, with 1 indicating that the answer described something not aligned with the current status of the PRU and 5 indicating full alignment.

Finally, the Excel spreadsheets were examined and summarized. Again, citing the example of the PRU staff, all six interviewees had marked the following factor as important for PRU functioning: "The staff consists of professionals with different types of education, knowledge, and competence (within education or social work), contributing different types of education, knowledge, and competence (in pedagogy or social work)". They had rated this factor as one of the eight most important. The interviewees also rated to what degree this factor was aligned with the current state of the PRU; three interviewees rated this as 5 (indicating full alignment), two as 4, and one as 3 (this example comes from the "Results" section; see Table 1).

**Table 1.** Statements regarding which factors are important for good PRU functioning, agreement about which factors are most important, and individual ratings as to whether the statements correspond to the state of the PRU together with group averages.

| Items (Answers that Are Formulated as Statements) | Agreement | Average Rating | Individuals' Ratings | | | | | |
|---|---|---|---|---|---|---|---|---|
| **Current students** | | | | | | | | |
| to be in the class in the homeschool often (at least once a week). | 2 of 3 | 2.0 | 1 | 4 | 1 | | | |
| school work is given higher priority (there are too many other activities). | 2/3 | 2.0 | 1 | 1 | 4 | | | |
| **Former students** | | | | | | | | |
| sufficient with resources to have a teacher - student ratio of 1:4. | 4/5 | 4.8 | 5 | 5 | 5 | 5 | 4 | |
| **Legal guardians** | | | | | | | | |
| the communication between legal guardians and the PRU staff works well. | 5/6 | 5.0 | 5 | 5 | 5 | 5 | 5 | 5 |
| **Staff at the PRU** | | | | | | | | |
| the staff consists of professionals with different education, knowledge and competencies (within education or social work). | 6/6 | 4.3 | 5 | 4 | 5 | 4 | 5 | 3 |
| **Principals for regular schools** | | | | | | | | |
| the responsibility for documentation is clarified and follows educational legislation (common routines). | 4/5 | 2.6 | 3 | 3 | 2 | 2 | 3 | |
| **Chief officers of the health and family sectors** | | | | | | | | |
| the staff working at the PRU wants to be employed there (own decision). | 2/2 | 5.0 | 5 | 5 | | | | |
| the PRU always works toward getting the student back to the home school. | 2/2 | 4.5 | 4 | 5 | | | | |
| the PRU staff have the right competence (teachers with teacher training qualification, social pedagogues with social work qualification). | 2/2 | 4.0 | 3 | 5 | | | | |
| the collaboration works well between the home school principals and the PRU principal (PRU students' feeling of still having their regular placement in the home school). | 2/2 | 3.5 | 4 | 3 | | | | |
| **Chief education officers** | | | | | | | | |
| the PRU staff work with solution based pedagogy and with high expectations. | 3/3 | no ratings | | | | | | |
| the work in the PRU is followed up and evaluated regularly. | 3/3 | no ratings | | | | | | |
| **Local school politicians** | | | | | | | | |
| the PRU staff has the right educational qualifications and work together to meet the children's needs. | 3/4 | 3.8 | 4 | 2 | 4 | 5 | | |

Most of the group interviews lasted three hours with a 20-min break, although the focus group interview with the current PRU pupils lasted only one hour. The individual interviews were also shorter, lasting less than an hour. There were also several shorter conversations with people (e.g., administrators) who had worked or were working in positions that had some weaker connection to the PRU. During the observation days, I had frequent conversations with PRU students.

All individual and group interviews (including focus groups) were transcribed verbatim for analysis (302 A4 pages of text).

### 4.4. Data Analysis

A stepwise analysis of the interview data was conducted. The first step was to sort passages from each interview transcription into six categories corresponding to the research questions. Four of these categories were informed by the IRIS model (Figure 1), while the other two were specifically formulated for the PRU studied here. The four categories informed by the IRIS model are policy related to the establishment and running of the PRU, special education provided and/or required at the PRU, academic and social outcomes for PRU pupils as well as other outcomes closely related to the PRU as an educational institution, and experiences related to the PRU. The two categories specifically formulated for the PRU are general descriptions of PRU students and of the reasons for their placement and answers indicating potential alternatives to having a municipal PRU. Text reduction followed in which the most relevant interview material for the six categories was presented in the form of condensed descriptions retaining the original wording as much as possible. All descriptions were indexed for control purposes, making it easy to find and reread them in the interview context as necessary. These 327 descriptions were positioned in A3 format matrices screened into the different stakeholder groups and six categories to enable comparisons. This data arrangement provided an overview of the groups, illustrating to what extent the focus groups had the same or different opinions, whether or not the participants in a certain focus group agreed within a category, and whether the interviewees were consistent in their answers. Within-group similarities and differences between the interviewees were also examined based on each interviewee's choice of what was important and on the degree to which each statement was aligned with the current state of the PRU according to the interviewees' perceptions. The aim was to examine the dominant perceptions in each focus group as well as the variation among interviewees in each group, as both are important for a nuanced picture of the material. Based on the second aim of the study, i.e., to compare the students' experiences with those of the other stakeholders, the current students' descriptions are paid more attention in the "Results" section than the other stakeholder groups' descriptions.

### 4.5. Trustworthiness

Focus group interviews were the main data collection method, as they are useful for exploring experiences and promoting insightful discussions in which interviewees are likely to express their true views [24]. Several measures were undertaken to enhance the quality of these interviews and other parts of the study. These measures will be described in relation to credibility, transferability, and dependability [25]. To ensure credibility, thorough preparation was undertaken, including training in the focus interview format used here [23] on three occasions and a pilot study to test the efficacy of the research instruments [26]. The preparation also included conversations with three researchers who had extensive experience of conducting different types of focus group studies and one workshop with a research group whose members use the focus group interview technique. The format used required strategies and techniques differing from those of a standard interview. For example, strategies had to be worked out in advance concerning where the researcher should sit, how the group should be positioned around the conference table, and guidelines on the role of the researcher acting without a second researcher. A second researcher is often used in other types of focus group interviews but was not needed in the approach adopted

here. As part of the preparations, a pilot study was conducted with two participants, in which all procedures in the focus group format were tested and the participants had to give feedback. The handling of the technical aspects had to be automated, so that the interview technique would function during the interviews. To ensure transferability, this article presents thorough descriptions and carefully chosen quotations from interviewees. The quotations represent the interpreted gist of the interviewees' descriptions to help the reader understand what the interviewees were trying to convey. Regarding dependability, the aim was to thoroughly describe the interviewee selection, data collection, and analysis of the collected material.

## 5. Results

The presentation of the interview results follows the order of the research questions. Quantitative data from the focus group interviews and observation data are presented after the results.

### 5.1. Description of PRU Students and Reasons for Their Placements

The analysis found differences between stakeholder groups concerning the perceptions about what students should have their placement in the PRU. The students, both former and current, described having learning difficulties. In both groups, some interviewees described previous schooling in which they had been bullied and had responded physically. An additional reason for placement was their social circumstances, which, according to the former students, had a large impact on their life situation. One student disagreed, saying that the social service sector had never been involved. Neuropsychiatric conditions were not mentioned by the students. It was, however stated by a few legal guardians that the PRU group also contained one or more students with a neuropsychiatric condition. Participants in several other stakeholder groups had the opinion that students with a neuropsychiatric condition usually consisted a part of the PRU group. In these descriptions, two positions were evident: that there are social conditions that at least partly underlie the school problems and that current PRU students differ from the former students from ten years earlier. The reason that students receive a PRU school placement is more often due to neuropsychiatric difficulties today, than it used to be. A common opinion was that the current students more often have neuropsychiatric rather than social problems and therefore need medication. In follow-up questioning, no one could explain this change, while others claimed that the students were still of the same type. Most descriptions were of overt aggressive behaviour. A few mentioned "disturbed behaviour", and several different stakeholder groups were obviously describing the behaviour of the same child as the descriptions were strikingly similar. The child had had a PRU placement several years earlier. Regular schools often responded to the students' inappropriate behaviour by segregating them. One legal guardian described the school situation before the PRU placement: "He had private sessions alone with his assistant for one and a half years, and the assistant was dyslexic and was required to teach him all the subjects". There were more descriptions in the focus group interviews of similar ways of arranging the school situation for students. The PRU leader commented on previous work in a regular school with a student who just had been placed in the PRU. The leader said that it was very strange that so much special needs provision over many years appeared to have had no effect at all.

### 5.2. Policy

There was almost complete agreement that the most important task of the PRU is to return the students to their home schools. According to many responses, the strategy is to remove the students from school situations where they did not fit in or enjoy their schooling, support them in the PRU, and work with the family in collaboration with Social Services. This is described as "taking them off track" for a while. Several stakeholder groups described this as a way to improve these students' schooling so that they would earn passing grades and have an educational setting in which they feel comfortable. An

alternative view was expressed by the current PRU educational leader, who stated: "A goal must be that there is no PRU". The leader added that the regular schools were inadequate and could not provide a conducive setting for all their students.

Both student stakeholder groups claimed that they were forced to leave their home schools and study at the PRU. According to the current students, the PRU is a school setting characterized by rules where they are always being watched and guarded. Two of them claimed that the teaching there was lesser in quantity and quality than at their home schools, meaning that they would never catch up with their former classmates. They were upset at this, as the main reason they were given for PRU placement was that it was required so that they could catch up with their home school peers; if they did, they could return to their home schools. The former students' descriptions were different. They described the PRU as a type of school where they received a lot of support as students. They did not say whether the goal of their placement was that they should return to their home schools or whether the intention was to continue at the upper secondary school level. The latter was likely the case for two of them who were placed in the PRU late, in ninth grade (the last year of secondary school). Some student stakeholders described how their legal guardians had demanded PRU placements for them.

The view that the PRU should provide good-quality education and have special competence that was unavailable in regular schools was shared by most stakeholder groups. There was also agreement that other factors were decisive for establishing the PRU, which was started for economic reasons according to the politicians, chief education officers, and the chief officers or the health and family sector. The cost of placement in institutions offering treatment had increased greatly, and the PRU was supposed to reduce such expenses. These stakeholder groups emphasized that it was less stigmatizing to attend a PRU than other treatment institutions. However, one of the chief officers of the health and family sector also emphasized that placement in a PRU is a "very big step in segregation".

Many stakeholder groups discussed a previous report by the School Inspectorate that criticized the PRU education. The criticism was strong concerning several aspects of work at the PRU. One claim in the report was that the students were not given supportive conditions to obtain grades in all subjects.

*5.3. Provision*

Whereas the current students acknowledged that the PRU gave them support, they also criticized their schooling, saying that too much time was spent on activities outside the PRU. They described these activities as hindering them from catching up with students in regular schools. The former students described their schooling at the PRU in more positive terms. They emphasized more than the current students that they received a lot of support, they all felt accepted and in a relationship with the PRU staff. The PRU staff was also persistent and never gave up in their effort to support, the staff had patience and they always remained calm, understanding and motivating their students. This differed from what they had previously experienced at their regular schools. Included in the motivational work, according to the students, was that the PRU offered breakfast:

"That was wonderful!" (Girl 1)

"Yes!" (two boys in chorus)

"That made you come on time—you couldn't miss breakfast at school". (Girl 1)

According to the students, token reinforcement programmes, within a behaviour modification framing, were used. This was useful for some students, as a conversation with another girl showed:

"If you behaved in an orderly way during the week, you got a reward on Friday".
(Girl 2)

"Yes." (two boys in chorus)

"Why was this important?" (interviewer asking for clarification)

"Then you felt that you had managed it in some way". (Girl 2)

The legal guardians, especially those representing the former students, agreed with their children's descriptions. All legal guardians of the current and former PRU students agreed that communication between them and the PRU staff functioned well, as they felt invited to collaborate in their children's schooling. All adult stakeholder groups emphasized that high-quality relationships between families and the PRU were important. This was often declared to be the most important aspect of PRU collaboration, with the collaboration with social workers in the school health sector also being stressed as important. The former students mentioned this collaboration indirectly or only stated that they, with one exception, had contacts with municipal Social Services. The current students did not mention such collaboration or contacts at all. The former students confirmed that PRU staff had come to their homes to encourage them to come to the PRU when they were absent without permission; this part of the support was also emphasized as important by the principals of the regular schools and the chief education officers.

Many stakeholder groups comprising adults, including the legal guardians, argued that it was important to "take the students off track", i.e., take them away from the regular school setting and offer them support at the PRU where the staff density is much higher. The legal guardians based their arguments on previous experiences of schools that could not or were unwilling to provide their children with schooling that worked. This was largely described as insufficient competence in regular schools regarding neuropsychiatric diagnoses. A different opinion was expressed by one legal guardian who works with people who have neuropsychiatric diagnoses:

"It is too simple to say that this student has ADHD or Asperger's".

"Yes." (another legal guardian agreed)

"These simplified labels that we have today, I work with them and I know how easy it is to put children into these categories. Then they never see anything else at school and, regardless of what they do, they are marked".

When asked for clarification about neuropsychiatric diagnoses and what teachers should do based on them, no clarification was given by any of the legal guardians who had emphasized the importance of neuropsychiatric knowledge. One chief officer of the health and family sector said: "We have taken a look at the children who come to the PRU and tried to develop staff competence based on that, so we have offered a 7.5-credit course on neuropsychiatry to some and we are now looking for this knowledge when we hire new staff".

No specific teaching method that could be regarded as neuropsychiatrically based was described in any interview. Some interviewees said that structure and rules were important. An individualized teaching approach and "solution-based pedagogy" were emphasized as important by most adult interviewees in the stakeholder groups. "Solution-based pedagogy" was described by PRU staff as "working on the things that work (i.e., in which the students function well) and not focusing on the problems they have". One chief education officer said: "This is so basic and instinctive to me and I have not had any formal training in this". Nearly all stakeholder groups assumed that the teaching should be individualized at the PRU. One leader of the school and health department added: "They have their teaching separated from other children to a very large degree—they are not group-oriented children".

The PRU had seven full-time staff and often only seven students with a maximum of ten students. Most stakeholder groups deemed it important that the staff be highly competent, having university qualifications and considerable experience working with students needing special support and living in difficult social circumstances. One principal from a regular school said: "If we should say what would be really important, they should be super pedagogues, and with a lot of experience, of course." The PRU school staff maintained that the PRU could not succeed with students who needed a school offering treatment; these students were eventually placed in institutions.

*5.4. Outcomes*

One current student told, with what seemed like envy, about one student who had succeeded greatly two years earlier, leaving the PRU for placement in a regular school. The student said: "I am probably one who won't succeed". The former students continued at the upper secondary level, even if they did not qualify for national programmes but rather for preparatory programmes. There is agreement among them that the support in the PRU helped them to finish the nine-year compulsory schooling. The parents of the former students were satisfied that their children had received passing grades in more subjects than before, and one parent of a current student expressed similar satisfaction.

The descriptions from all other stakeholder groups were similar. Nobody knew about the overall outcomes of student groups placed in the PRU or what happened to them after they finished their schooling in the municipality. There were anecdotal indications of good progress for previous PRU students, mostly concerning only one student in the municipality. Furthermore, it was known that the PRU could not serve the group of students for which it was originally established, so the costs of student placements at treatment institutions had not decreased. Concerning the PRU, there was no documentation, no quality assurance work or reports on such work, and no evaluations or follow-ups. As one chief officer of the health and family sector said, "We don't have instruments to see whether we have succeeded with the PRU", and one politician said: "It is annoying to have to admit to you that we don't have 100 per cent control".

*5.5. Experiences*

Two current students had lost contact with their former classmates in their home schools. One student was very upset about this, especially about losing contact with his closest friend since preschool years. The distance from the PRU to his home school had made it almost impossible to have part of his education at his home school. His friend had a lot of homework and a school week schedule that was different from that of the PRU student. Taken together, the PRU placement had made it almost impossible for them to stay in contact, according to the student. Another student's home school was much closer to the PRU, so he had never lost contact with the home school and his classmates there. He had been able to continue having some lessons there, as he could easily move between schools during the day. He said that the other students, due to the long distances to their home schools, had a much tougher situation and that he was quite pleased with his own situation.

The former students described several problems with their schooling at the PRU. Being associated with the PRU was one. Some of the students had been threatened by regular school staff that if they did not behave properly, they would be sent to the PRU. One student said that he was viewed as similar to a jailed prisoner by regular school staff, due to his placement in the PRU. He repeatedly raised the issue of how the PRU was regarded by others. He said, for instance, that he was ashamed of being connected to the PRU, and that when the PRU group met other students outside the PRU he "wanted to take off his head and kick it away, to stop seeing everything" and that he was ashamed of being a difficult child. Another male student agreed, while a female student said: "It wasn't that way for me—I only hung out with other difficult children". These students also laughed about the rule that they were forbidden to visit their home schools but not when the class photos were taken: then, their home schools insisted that they should participate.

The legal guardians said that they had had a difficult time when their children were in their home schools (i.e., regular schools), and they felt hurt at having been questioned by school staff. One legal guardian said that they (the legal guardians) had "received a call from the school, which I was so scared to death of for a while, that I could not answer the phone". The legal guardians compared the different attitudes at the home schools and the PRU. The home schools conveyed blame and the indirect message that they should come and remove their children from school but signalled no intention of seeking solutions. The PRU, in contrast, always tried to find solutions and did not dwell on what had happened.

This was called a solution-based approach by the legal guardians. They also praised the PRU for contacting them and giving positive messages about their children's development.

School staff were mainly positive regarding their work at the PRU and their collaboration with the legal guardians and other parties, such as Social Services and the home schools. However, they described the home schools as differing from one another, with some principals wanting to collaborate, while others did not. Some of the home schools were unwilling to let the PRU students return. The educational leader of the PRU stated that the home schools had to change, as they were struck in old structures and modes of thinking. Both the current and former PRU leaders echoed the PRU staff, describing different attitudes among home school principals, some of whom were willing to collaborate while others were not.

This issue of enabling the students' return to their home schools was intensely discussed by the principals in their focus group interview. There was agreement that it was very difficult to welcome the PRU students back to their home schools. One principal described a hypothetical situation facing a principal in which a bonus system could be called for, which resulted in a lot of laughter mixed with serious discussion among the principals:

> "I think it's like this, I think it's good that 'he' has done schooling at the PRU for five years and that I only pretend that I want that student back" (i.e., the main argument for having a bonus system, suggested by this principal).

> Another principal replied: "I think it's our damn responsibility and mission to get them back to their home school as soon as possible".

> The principal who suggested the bonus system clarified: "But I don't think that will happen".

The chief officers of the health and family sector criticized the employment policy at the regular schools, which they thought was wrong and had to be changed. One chief officer said: "We hire young people who have just finished upper secondary school, and they may be employed as resource persons for the children with the greatest difficulties. The municipality must rethink this, as we are not providing the children with what they need".

### 5.6. PRU Alternatives

Several stakeholder groups discussed the idea of a separate school for the PRU students. The establishment of a separate school, a PRU, was decided by the politicians based on the suggestion of one employee in the municipal central administration. This person had worked in institutions for children in need of treatment. The suggestion was aligned with the economic problems of the municipality, as the costs of institutional placements had increased too much. Some interviewees in the stakeholder groups favoured meeting the students' special needs in their home schools, whereas others favoured placing the students separately from their classmates. When one politician emphasized that the PRU students still would have contacts with students from other schools, another one replied: "Yes, but they will still be in an enclave".

Several stakeholder groups cited the example of a principal who had improved the special pedagogical competence in the home school based on an inclusive education policy. According to the politicians, the school succeeded so well that, without any extra resources, PRU referrals were no longer needed. The principal gradually developed the teacher competence over a long time to realize inclusive schooling. Then, the principal left the school and the municipality for a higher position in another municipality. The two politicians, who had been involved in establishing the PRU ten years earlier, described the case in detail, and one of them summed up: "What he did succeeded as long as there was a shared policy, but when he left it started to crumble, and such systems (systems that are depending on one person's leadership) are not good".

*5.7. Factors of Most Importance Regarding the PRU According to Stakeholders' Ratings*

When the interviewees in the stakeholder groups answered the question about what was important for good PRU functioning, many different factors were suggested. The number of suggested factors varied between stakeholder groups, as mentioned above. The current students, who suggested only eight factors, were asked to rate the three most important factors for good PRU functioning, which is close to a third of the suggested factors. The chief officers of the health and family sector chose eight factors—one third of their suggested 23—rated as especially important. In this section, the main focus is to compare what factors were considered the most important by each stakeholder group. This is based solely on the factors that received the most votes in each group, which is a methodological choice that may be questioned. However, a result presentation of more factors from each stakeholder group would result in a large number of factors making comparisons very complicated as all factors were unique in their wording. As seen in Table 1, between one and four factors were rated as particularly important by each of the various stakeholder groups, for a total of 13 important factors.

In the group with the four current students, one student had left when it was time to choose which three factors were most important. That student, who did not seem to get along with two of the other three students, wrote the most important factor on a separate paper: "It's more peaceful during lessons so I get more done. I had more friends at my home school. I have received passing grades in more subjects at the PRU". Two of the three students who stayed for the whole focus group interview voted that "often being in class in the home school" was among the three most important of the eight factors that the group had suggested.

All participants also rated to what degree each factor was aligned with the current state of the PRU. Concerning the factor "often being in class in the home school", the ratings given by all three current students on the scale of 1–5 were low at 1, 1, and 4, which gave an average rating of 2.0. This means that, according to two students, the PRU did not live up to what the group valued as important, whereas the third student apparently had been in class in the home school quite often, hence the rating of 4. Their ratings differed regarding the next factor, which the first student again gave a low rating of 1, but now the second student gave a high rating of 4 and the third student assigned a rating of only 1. This underlines what the three students said during the interview and further reveals differences within this stakeholder group.

The within-group differences were less pronounced in most other stakeholder groups, depending partly on differences regarding the content of the identified factors. Two chief education officers did not want to rate the extent to which the work or conditions in the PRU corresponded to some of the factors, including the two in Table 1, so individual ratings have been omitted for that group. The ratings show, in line with the transcribed interview data, that in most stakeholder groups, there was agreement about what was most important for the PRU to work well. Their individual ratings were also usually in agreement regarding the factors rated as most important by each group as a whole but there were clear differences between the groups as to what was deemed most important. The goal of returning the students to their home schools, was expressed by the current students in two factors, keeping contact with their home schools and schoolwork prioritized so the students could catch up with their classmates in their home school. Notably, this goal of returning the students to their home schools was the top priority in only one other group, i.e., the stakeholder group of the chief officers of the health and family sector. Of the 142 unique factors identified in all stakeholder groups, this one is only found in one more stakeholder group, namely, the group of politicians, in which two of the four politicians rated this factor as one of the most important.

## 6. Discussion

The experiences and views of current PRU students are of great importance in this article, as their voices are often ignored in research. It is especially important in the

Swedish context to enhance our knowledge of how these students perceive their situation in PRUs, and this study was intended to accomplish this. In the following discussion, the current PRU students' comments will be compared with those of other stakeholder groups. This discussion will be based on mainly the responses of the three current students who completed the whole focus group interview.

### 6.1. Limitations and Strengths

This study has certain limitations, one being that its results come from only one PRU; however, as the purpose was to contribute sound knowledge based on information from many stakeholder groups based on one PRU, this limitation has been taken into account. Moreover, the interview sessions were long and contained many clarifying discussions. Data saturation occurred at each focus group session when the interviewees could no longer contribute statements describing new factors. Based on the chosen interview format, all interviewees' voices were individually documented, except when one current student left the interview. A great many data were collected, based on individual interviews, other conversations, school documents, and observations that were intended to give the researcher contextual knowledge before the interviews. This thorough data collection is a strength of this study, and the theoretical framework provided a useful analytical tool with which to explore the large amount of data.

### 6.2. The PRU Is Ideal for Whom?

There were different opinions about the PRU establishment. According to the stakeholders who started the PRU, it was started based on an economic rationale. The intention was to reduce the expenses associated with the original target group, i.e., students placed at treatment institutions. This coincided with legal guardians' demands that the municipality should provide better educational conditions for their children. A local solution was suggested, and the PRU was started. This arrangement did not work, according to the interviewees, as the PRU could not offer a solution that worked for the target group. Interestingly, once started, the PRU became popular, but the target group changed. The consequence was that, instead of improving teaching quality in the regular schools, a new segregated school institution was established.

The PRU clearly has a function in the municipality when regular schools are not equipped to address a few students' inappropriate behaviour. There are three types of such behaviour in students: withdrawal, acting out, often combined with aggression, and behaviours described as abnormal and difficult for staff in regular schools to understand. Children manifesting the last type of behaviour are so rare that the examples of such behaviour discussed in several interviews likely all relate to a single case. Regarding students' descriptions of their own behaviour and its causes, they cite external factors such as having been bullied. They attribute their behaviour to external factors and reveal a lack of perceived responsibility, according to Jalali and Morgan [11], who apply a psychological perspective in analysing student behaviour. For the students in the present study, psychological explanations seem too simplistic. The students' social background is one other factor, and Social Services have been involved in most of the students' families, recalling the students studied by Hart [13]. Most of the students had experienced social circumstances that must have affected their wellbeing at least during some period over their schooling. These students' situations seem to have been exacerbated in the regular schools by the special support they were provided to meet their needs. The interviewees cited many examples of students having been provided with educational conditions of very low quality. According to the interviewees, this especially affected students from socially disadvantaged families. Interestingly, one chief officer of the health and family sector seemed to downplay the relevance of the social background of the PRU students. This chief officer said that children with neuropsychiatric diagnoses have replaced the earlier PRU students to a large extent. However, the social difficulties experienced by the current PRU students' families seem serious. Furthermore, it seems highly improbable that social

circumstances could have improved so dramatically over a ten-year period. That several students over the last years also may have had medical diagnoses can partly be explained by the current over-diagnosis of students [27,28]. It does not seem justifiable to reduce the three interviewed current students' life circumstances, or other recent PRU students', to individual-level factors such as their use of maladaptive coping strategies [11] and/or their biological constitutions, as in neuropsychiatric explanations, as suggested by the conversations, interviews, and other utterances made by participants in the focus groups. If such factors are dominant, why does the PRU focus so strongly on working with Social Services and the families? The first research question, concerning the underlying problems making PRU placements necessary, is not easily answered. The official underlying rationale of the PRU was to provide a supportive school setting for students with special needs that regular schools could not meet, with the aim of returning these students to their home schools as soon as possible. The interviewees described different student target groups and different causes of their need for special support. It is clear, however, that the PRU was originally established for a target group that it could not handle: a very small number of students with extremely complex life circumstances. The idea of taking them "off track" for a short while and then returning them to their regular schools appears naïve.

*6.3. Stakeholder's Views of the PRU*

Research questions 2–5 focusing on policies, provision, outcomes, and experiences of PRU placements will be addressed here. PRU policies were well-known by the stakeholders, the main policy objective being to return the PRU students to their home schools. The three current students said that this objective was very important for them, in line with Jalali and Morgan's [11] findings. For this to happen, the students first had to be segregated from their home school—taken "off track", as it was described. Logically, this would seem to be a detour, and this is the strategy also used in the UK [11,13]. As the results showed, only one of the three current students was satisfied with his contacts with and visits to his home school, and he was confident about being able to return. The situation differed for the other two current students, who were frustrated and discontented with three things in particular: first, they were given no choice about their school situation; second, they had almost completely lost contact with their home schools; and third, they had been promised that the PRU education would help them catch up with their home school peers, although this proved not to be the case. It appears that these students had analysed their situation correctly, as these problems were well-known in other stakeholder groups and made it very difficult to return the students to their home schools. In this way, establishing the PRU increased inequity problems in the municipality. Furthermore, the policy that the PRU students should return to their home schools was not supported by all stakeholders. The interview with the principals, for instance, revealed two antithetical positions: one was that it was not self-evident that PRU students should return to their home schools, and the other was that it was the home school's responsibility to take these students back. Interviews with other stakeholder groups and individual interviews support this description. The idea of a bonus system to incentivize the principals' efforts to enable student return to their home schools indicates that this policy objective also concerns issues of resources and responsibility. Interestingly, few stakeholder groups identified this main policy objective as an important factor.

The geographical area of provision is closely connected to the main policy concerning the PRU. The construction of one PRU geographically distant from some of the regular schools was a great barrier impeding two of the three current students from maintaining contact with their home schools and with friends and from eventually being reintegrated. Another barrier perceived by the three students was the teaching in subjects necessary for success in school. They perceived such success as required for reintegration and expressed disappointment with the time spent on other, non-scholastic activities. The School Inspectorate report supports them in this criticism. The observed teaching was delivered by engaged teachers, but the teaching was conventional and based on the same

pedagogical principles used in most regular school classrooms. However, the PRU teaching was mainly delivered one-to-one or, occasionally, one-to-two. Compared with the regular-school teaching described by the stakeholders, in which students were separated from the class for a long time and taught by student assistants lacking education for this work, the PRU teaching was obviously much higher in quality. It is interesting that the resources and ambitions were so much higher in the PRU. The stakeholder groups comprising professionals responsible for educational quality emphasized that the staff working with the PRU students must be highly competent. Suggestions that staff should have university-level qualifications and advanced experience to work at the PRU met with approval. This difference in ambition level, which was never explicitly discussed in any stakeholder group, is striking and difficult to understand. To answer the third research question about what support the PRU provides, is it evident that substantial resources created a better learning situation for the PRU students, especially compared with what they had experienced previously. Nevertheless, several quality issues were reported by the School Inspectorate.

The fourth research question about the PRU outcomes according to the stakeholders can be answered briefly: the current students' outcomes were to come in the future. The current student who participated in only part of the focus group interview wrote about achieving passing grades in more subjects. The former students had not qualified for the national programmes for which 80–85 per cent of Swedish youth qualify. They had, however, continued with their studies at the upper secondary school level but on introductory education for students who do not fulfil normal eligibility requirements. Note that a comment was previously made about the former students and their legal guardians, who do not appear to be representative of legal guardians in general whose children attend the PRU. Interestingly, anecdotal descriptions dominated discussion of the PRU students' outcomes. In fact, little systematic data collection, follow-up, and evaluation of the PRU were conducted in the municipality, and the politicians expressed embarrassment at this. The fourth research question about PRU outcomes therefore cannot be answered in a valid way. This has been a common problem regarding PRUs in general, being reported in one total population study [9] and recently described as problematic by a Swedish commission that recommended measures to improve the situation [3].

The stakeholder experiences naturally differed both between and within the groups, and several stakeholder experiences have already been discussed in relation to the areas of policy, education provision, and outcomes. Other mainly negative experiences related to being a PRU student will also be discussed here. Across the stakeholder groups, there was awareness that the segregation of the PRU from the home schools is problematic. One chief officer of the health and family sector said that it was a "very big step in segregation" and expressed concern about such measures. Results from other PRUs are that stigma was associated with being transferred from a regular school to a PRU [29]. Several participants in this study reported that school staff threatened misbehaving students with transfer to the PRU, and one student said that, as a PRU student, he was regarded as a prisoner. One student expressed strong feelings of shame at being a PRU student. In answer to the fifth research question about experiences of PRU placements, it is evident that PRU placement may negatively and substantially affect the situation of students who have already experienced several perceived school failures. Based on the participants' descriptions of the negative consequences of segregated schooling [3], the question is whether there are any alternatives.

### 6.4. Alternatives to the PRU

The sixth research question concerns whether the stakeholders believe that there are alternatives to PRUs. The idea of establishing a PRU entirely separate from the regular schools was based on a suggestion from one person in the municipal educational adminis-tration. This person, who had worked in an institution for students needing treatment (the PRU's initial target group), suggested establishing a separate school. The decision to do so was discussed in several groups, and some of the stakeholders expressed the opinion

that the resources for special needs education would be better used in regular schools. This would appear to be an alternative that might reduce the risk of stigma. Several stakeholder groups discussed the example of a school whose principal had increased the staff's special pedagogical competence so that the school would not need to refer students to a PRU; another case of such a regular school was reported in a previous study by the present author [2]. Interestingly, those schools increased their capacity to create what seem to be inclusive learning environments without requiring any extra resources. Accordingly, the answer to the sixth research question is that there are alternatives to PRUs, some of which may reduce or even eliminate the demand for PRU placements.

### 6.5. Contributions of Stakeholder Theory

Stakeholder theory was used to identify study participants (Figure 2) and, during data collection, to identify additional relevant stakeholders who had not yet been recruited. Several other stakeholders were mentioned by the interviewees, such as students and staff in the regular schools, Social Services, and, in particular, the School Inspectorate. Stakeholder theory has brought an ethical dimension to business research [18], broadening corporate responsibility to encompass more stakeholders than just the shareholders. It has been recognized that some stakeholders lack power [30] and that their only choice is to trust their organizations. From an organizational perspective, some stakeholders who participated in the study lacked power, whereas others occupied strong power positions. The PRU students and the legal guardians described many poor-quality educational situations in their previous regular schools. Transfer to a PRU may have been the only way for these students to obtain a better education. The lack of systematic follow-ups and evaluations, however, makes it impossible to determine whether the PRU represents a good educational solution for future students. Again, it should be recalled that the groups of former students and legal guardians of former students were not viewed as representative of other students and their legal guardians in this study. Nevertheless, many students and their guardians stated they had long experienced difficulties and had often been blamed by others, including staff in regular schools. In relation to the municipality and schools, the guardians were stakeholders with low self-confidence and little power, according to many participants. As neither the municipality nor the PRU fulfilled their responsibility to document the PRU work, the School Inspectorate appeared to be a strong stakeholder with considerable power. The School Inspectorate claimed to uphold the students' right to a good-quality education, which they had not received.

## 7. Conclusions

An increase in the number of PRUs has been recommended in Sweden, as has previously been reported in the UK [31]. There is no scientific support for the premise that Swedish PRUs offer improved conditions to the students placed in them. In Sweden, the dominant official discourse is that students experiencing severe school difficulties will receive better education if there are more PRUs dedicated to serving excluded students. The official discourse has been the same in the UK [32]. However, another strong motive for establishing PRUs is to remove disruptive students from mainstream schools, as their presence has been described as hindering other students' education and the improvement of schools [32].

Disruptive students seem out of place in the current neoliberal era, with economic steering models based on New Public Management and demands for efficiency and higher-quality public education based on market models. In this context, there is demand for segregated settings and medical labels for students in need of special education. The focus on individual deficiencies de-emphasizes causative factors based on systemic societal conditions [33]. The situations of some individuals may even worsen in PRUs, as "PRU attendance may actually contribute to the presence of mental health problems" ([11], p. 64). There is an urgent need to further elucidate the school and life circumstances of students who experience severe school difficulties and who risk being referred to PRUs.

This requires that their voices be listened to by stakeholder groups with the power to decide on the objectives of the school system and of society more broadly.

**Funding:** This research received no external funding.

**Institutional Review Board Statement:** Ethical review and approval were waived for this study as it was conducted according to ethical regulations in Sweden, where Ethics Committees at universities and university colleges are rare and where ethical approval is managed by the Swedish Ethical Review Authority. Ethical approval is required when sensitive personal information is registered or handled, for instance information that reveal an individual's religion. The material for this study was collected in an agreement between Jönköping University and one municipal, and it does not contain sensitive personal information. Furthermore, Jönköping University does not have an Ethics committee.

**Informed Consent Statement:** Informed consent was obtained from all subjects involved in the study.

**Data Availability Statement:** Not applicable.

**Conflicts of Interest:** The author declares that there are no conflict of interest.

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
