# Peer review of "The PRU: The Solution for Whom?"

_education, doi:10.3390/educsci11090545_

Round 1

Reviewer 1 Report

This paper addresses a very interesting topic that is relevant beyond both Sweden and the phenomenon of PRU provision more specifically. In fact, it is arguable that its most interesting findings concern the Swedish education system more generally rather than the use of PRUs in particular. It also presents a very strong discussion of the study's findings in the Discussion section. This aspect is most interesting.

However, the introduction and overview of the literature need revision and focusing. For example, at no point did I see the PRU as a concept defined or explained coherently. This is a significant omission given its centrality to the overall paper. In addition, the literature review and its focus on the increased use of PRUs in Sweden in recent years could have been improved through being contextualized in the international movement towards inclusive practice in education. Why has Sweden moved towards increased use of PRU's as a form of segregated provision? Is there support for such movements in the existing literature or in other jurisdictions? What does this say about the challenges to the mainstream education system caused by attempts to develop inclusive practices within schools and increase the diversity within the pupil profiles in mainstream classrooms?

I like the use of the stakeholder theory-informed IRIS framework which is an interesting approach and holds much promise for exploring this complex phenomenon. However, this section reads like it has been taken from a larger and more detailed report where the author(s) may have discussed this topic more expansively and more effectively contextualizes this research. 

The methodology section is also missing important details required to clearly outline the methods used. What observations were collected by the research team? How was this done? Who was observed and how was the collected data analyzed in the context of the focus groups/ interviews? In addition, the section on how the overall data was analyzed is very brief and more detail would be required regarding how the findings from across the (often complex) data collection were brought together in the analytical process? A clear rationale for the novel and sometimes complex methods used within the focus groups would also have improved this section. This is stated but not explained. 

The findings are presented according to each research question, with the data from each stakeholder cohort presented in sequence. This is an unusual approach. What is striking is the increased detail and clarity presented within the Discussion section when compared to the previous sections. It is surprisingly more detailed and nuanced than the data presented in the findings section. The DIscussion is by far the strongest aspect of the paper and presents an interesting overview of the topic. 

The paper could be improved by focusing the introduction & review of the literature to be more in line and complementary in tone to the focus of the discussion section. In addition, a tightening of the method section such that it provides a more detailed and clear overview of the approaches used to collect and analyze the data would be adivsable.  

Author Response

Dear reviewer 1, Please see the attachment. /the author

Reviewer 2 Report

General comments

  • An essential study that analyses the impact of referring students with special educational needs for support and the implications as seen by the stakeholders.
  • Review of literature is well done and relevant.
  • In-text references need to have the year added. I do not know what referencing style is used. Some last names in the references are repeated like Jalali Jalali….Obert Obert, but not in all places. Needs to be checked for the right referencing style and uniformity.
  • Methodology is well narrated.
  • English and grammar need checking. I have marked in the first page but not in other places.
  • Under methods, the first para (lines 217 to 220) seems redundant. Instead, the method can be directly explained.
  • The results are well organised. It will be made more authentic if the direct quotes of concurring and contrasting views of the stakeholders on the issues also are discussed in context thus comparing the views to strengthen the argument.. It is seen in some places, but can be more elaborate in other places too.
  • Strength of the study is getting the direct experiences of the pupils in reference directly and i appreciate the authors for such an effort. This is a much needed study that will throw light on the issues related to inclusion, particularly the ones with emotional behavioural disorders which is less addressed so far. It will help the readers of this study to compare the status in their own countries  as the school system and political involvement differ between countries, though the issue is global.

Author Response

Dear reviewer! Please see attachment. /the author

Round 2

Reviewer 1 Report

The revisions have improved the article in line with feedback comments.